# Rowanberry—A Source of Bioactive Compounds and Their Biopharmaceutical Properties

**DOI:** 10.3390/plants12183225

**Published:** 2023-09-11

**Authors:** Ofelia Marioara Arvinte, Lăcrimioara Senila, Anca Becze, Sonia Amariei

**Affiliations:** 1Faculty of Food Engineering, Stefan cel Mare University of Suceava, 720229 Suceava, Romania; sonia@usm.ro; 2INCDO-INOE 2000, Research Institute for Analytical Instrumentation, 67 Donath Street, 400293 Cluj-Napoca, Romania; lacri.senila@icia.ro (L.S.); anca.becze@icia.ro (A.B.)

**Keywords:** rowanberry, nutraceutical compounds, phenolic compounds, biopharmaceutical properties

## Abstract

After a period of intense development in the synthesis pharmaceutical industry, plants are making a comeback in the public focus as remedies or therapeutic adjuvants and in disease prevention and ensuring the wellbeing and equilibrium of the human body. Plants are being recommended more and more in alimentation, in their natural form, or as extracts, supplements or functional aliments. People, in general, are in search of new sources of nutrients and phytochemicals. As a result, scientific research turns to lesser known and used plants, among them being rowanberries, a species of fruit very rich in nutrients and underused due to their bitter astringent taste and a lack of knowledge regarding the beneficial effects of these fruit. Rowan fruits (rowanberries) are a rich source of vitamins, polysaccharides, organic acids and minerals. They are also a source of natural polyphenols, which are often correlated with the prevention and treatment of modern world diseases. This article presents the existing data regarding the chemical composition, active principles and biopharmaceutical properties of rowan fruits and the different opportunities for their usage.

## 1. Introduction

The increased level of consumers’ awareness regarding safe foods that promote health—especially the search for new extracts and safe vegetal origin compounds that can be used in a versatile way by various industries in cosmetics, nutraceutical and pharmaceutical products production—raises challenges and scientific uncertainties [1].

Fruits and vegetables are considered to be healthy foods, mostly due to the presence of an abundance of valuable nutrients, such as minerals, polyphenols, vitamins, antioxidants, dietary fiber and more. Forest berries are regarded particularly favorably because, apart from the fact that they are one of the richest vitamin and photochemical bioactive sources, mostly antioxidants and polyphenols, they have distinct sensory characteristics and therefore are eaten fresh or processed: raspberries, wild strawberries, blackcurrant, blueberries, cranberries, cherries and so forth. However, many forest fruits remain underused, mainly due to their characteristic aroma, which is less pleasant for consumers, although their phytonutrients content is exceptionally high [2,3]. These forest fruits can supply various raw materials rich in certain compounds of nutritive and health interest [4].

The incorporation of a broader selection of fruits, particularly those currently underused, into production and consumption habits has the potential to greatly enhance human health, nutrition, livelihoods and ecological sustainability [4].

Mountain ash, also known as rowanberry (*Sorbus aucuparia* L.), is one of the underused species, similar to other related species [5]. *Sorbus aucuparia* is a species from the *Rosaceae* family, originating from the colder regions of the Northern Hemisphere, and is often found in high altitudes. Presently, it is used in wood production and ornamental horticulture and can be seen in gardens and parks [6,7]. It is appreciated for its nutrition and for its medicinal properties [8], especially for its bright red colored fruits with their functional properties [6].

## 2. History and Origins

The name of *Sorbus* was used before Christ, by Cato and Plinius, and comes from the Latin *sorbere*—to sip or to swallow, because the fruits and juice of several species were consumed in old times. It is believed to have magical power like mistletoe, and that its branches could be used to find treasures [9]. The Celts called it “the wizard tree”, and in Ireland, the tree played an important role in popular protection magic against spirits and the dead [10].

The fruits of many *Sorbus* species (*Sorbus aucuparia*, *Sorbus edulis* (Wild.) K. Koch/*Aria edulis* M. Roem., *Sorbus aria* (L.) Crantz, *Sorbus torminalis* (L.) Crantz, and *Sorbus domestica* L.) were added for use in food in various jams, confectionary products, syrups and forest fruits sodas [9,11,12,13].

In Perušić and Lovinac, the consumption of Sorbus species (*S. aria*, *S. domestica* and *S. torminalis*) has become an intriguing local tradition. The fruits were eaten as a raw snack in some towns from Croatia [14].

In a study conducted in Russia on wild dietary plants, *Sorbus aucuparia* had the most numerous proposed dietary usages, followed by *Rosa canina* L. and *Vaccinium oxycoccos* L. [15].

In the north of Greece, rowanberries were eaten due to their nutritive and medicinal properties [11]. Also, alcoholic beverages, such as wine, beer and spirit drinks, such as cognac or vodka, were brewed or flavored with rowanberries or their juice, mostly for the biochemical compound that helps cleansing and the preservation of alcoholic drinks. Additionally, rowanberries added flavor, astringency, bitterness and more sugar [11].

From a historical point of view, in popular medicine, rowanberries were used in ailments for gastrointestinal blockage and malfunction, diarrhea and for liver and gallbladder problems. They were also used for bronchitis and for their diuretic, anti-inflammatory, vasodilator, antidiabetic and vitaminizing properties [7,9,16].

Rowanberries were used for a long time in popular medicine as an appetite stimulator and a rich source of vitamins, especially ascorbic acid (vitamin C). They were also used as a juice for gargling and a gentle laxative remedy and a cure for rheumatism and kidney disease. The teas, syrups, jellies and alcoholic extracts from rowanberries were traditionally employed as remedies for flu, fever, infections, gout and rheumatism [16,17]. The fruits were traditionally used in Greece for their antidiabetic properties [11].

Rowanberries are recommended in popular medicine to treat hemorrhoids, high blood pressure, respiratory infections, rheumatism and gout. In Lithuania, rowanberry products were given orally for constipation and cough, while the bark decoction was used to wash the wounds, and apart from their effectiveness in cancer treatment, preparations made from rowan leaves were also used to address gastrointestinal problems and prostatitis [18].

In Estonia, the infusion made from the bark tree was used in cancer treatment [19]. In Romania, in homemade medicine, rowan was used against coughs, rheumatism, blood cleansing and tuberculosis. In pharmacy, the drug was known as “*fructus sorbi*” (*Baccae aucupariae*), with diuretic, antiemetic, astringent and antiscorbutic properties.

Products based on mountain ash are used mainly in traditional medicine for lowering cholesterol level, toning the nervous system, physical toning and delaying the aging process, visual acuity enhancement, in tuberculosis and rheumatism treatment, to calm coughing and for scorbutic treatment [20].

Almost all *Sorbus* plants parts have a historical recording of usage in different traditional medical systems, and in dietary, food, beauty, health and well-being products. The fruits were incorporated into pharmacies as polyvitaminic substances and included in daily food consumption. However, due to their distinctively bitter and astringent taste, they were often overlooked or infrequently utilized [4,5].

To surmount this challenge, sweet clones of rowanberries were initially chosen from the Sudeten Mountains (Czech Republic) during the 19th century. In the early 20th century, the Russian scientist and plant breeder Michurin initiated a program to enhance the sweetness of rowanberries in northern conditions. This effort resulted in the development of a fascinating group of hybrids of *S. Aucuparia* with other species: *Malus Mill.*, *Pyrus* L., *Aronia arbutifolia* L. *Pers.* and *Mespilus* L. [4]. Some of these hybrids have bigger fruits, with a darker color and a less tangy taste. Nevertheless, later investigations showed antioxidant capacity and bacteriostatic effect in wild rowanberry extract as well as in cultivated rowanberries [4].

*S. aucuparia*, rowanberry and other *Sorbus* species (*S. torminalis*, *Sorbus discolor* (Maxim.) Maxim., *S. domestica*, *S. aria*, *Sorbus chamaemespilus* (L.) Crantz, *Sorbus austriaca* Hedl. and so on) are typically cultivated as ornamental plants on a considerable scale in botanical gardens and park areas throughout Europe and North America [9,11].

Species of forest fruit-bearing trees play a vital role in preserving biodiversity by enhancing the genetic pool of the forest ecosystem, promoting strength and vitality throughout the forest and contributing to the improvement of soil quality. *Sorbus* are valuable species for softwood growth stimulation, after deforestation and for soil improvement [21,22].

The fruits are an important part of the diets of wild animals, especially birds, as they do not fall on the ground but remain in the trees [21].

Due to their climate and soil resilience, *Sorbus* plants are highly appealing for cultivation in standardized plantations. They can thrive in nutrient-poor soils, making them an excellent choice for such environments [5]. They are used in plantations to create green spaces and as ornamental species [20].

Encouraging fruit forest tree species, owing to their adaptability and resilience, can be seen as a highly promising endeavor to mitigate climate change. Planting these trees is an effective approach to restoring degraded forests and enhancing their ecological balance. Consequently, they hold significant potential for fostering sustainable development in rural areas, and their widespread adoption and utilization align perfectly with the principles of sustainable resource management [21]. The wood is used in carpentry, lathing and joining in wood making [20].

Different aspects of *Sorbus Aucuparia* are shown in Figure 1.

## 3. Botanical Description

The *Sorbus* genus has very complicated systematics, which include subgenera and sections that determine its diversity [23].

*Sorbus aucuparia*, named rowan, bird’s eye, Siberian mountain ash, Siberian keirn, cuirn and the Wiggin witch tree [6], is a deciduous tree which belongs to the *Rosales* order, *Rosaceae* family and *Maloideae* subfamily.

It is a slim tree that can grow up to 15–20 m tall in favorable growing areas, but it can present itself as a bush in poor soils. The tree has a round crown, with grey or red brown branches in its youth, which later become glabrescent, smooth and shiny. The bark is first smooth and grey, but later it forms a slim rhytidome and cracks into wide strips.

The leaves are imparted and compound, with 9–19 leaflets pairs and a shared petiole. They can be quite long, measuring 10–25 cm. From dark green, they become orange in the autumn.

The inflorescences are corymbed, multifloral and thick, with approximately 250 flowers. They measure 10–15 cm wide and are white, emitting a pleasant fragrance [22].

The fruits are fake drupe, globular, longer than they are wide, ovate, rarely ellipsoidal, 8–10 mm in diameter and often red colored but rarely red-orange. They stay on the tree in the winter, providing food for the birds. The seeds are narrow, elongated, reddish in, and number 2–3 inside each fruit [10].

It grows in rocky places, in well-drained soils, in deciduous or even coniferous forests, in peat lands from the mountain level to subalpine level and on sunny slopes. It vegetates well in fertile soils, with moderate acid humus, in the process of decomposition. It is a stress-tolerant species and is both frost resistant and shadow resistant. It bears drought well, if the drought is not prolonged [20,22]. This species is distributed across the entire Northern Hemisphere, ranging from low to high altitudes and extending from the Atlantic coasts of Europe to the Kamchatka Peninsula in Russia and eastern China [24,25]. The spread of *Sorbus Aucuparia* in Europe is shown in Figure 2.

## 4. The Chemical Composition of Rowanberries

The chemical composition of rowanberries is rich and varied, influenced by factors such as the location, origin, species, climatic conditions and ripening stage [26]. The phytotherapeutic and nutritional principles that can be detected in 100 g of dried rowanberries are shown in Table 1.

Glucose is the main sugar in rowan fruits [38]. Sorbitol is detected and quantified in rowanberry fruits [29]. As a natural non-glucidic sweetener, the fruits can be consumed by diabetics, although they contain glucose.

Rowanberry stands out as the fruit with the highest total analyzed organic acids content, in comparison with other fruits like: jostaberry, lingonberry, black currant, red gooseberry, hardy kiwifruit, vaccinium macrocarpon and aronia [28]. From the organic acids, the malic acid prevails in the composition of rowanberries [38].

Rowan fruits are rich in carotenoids [30]. Certain rowanberry varieties can be considered a rich source of carotenoids, the same as carrots [39]. From carotenoids, β-carotene—the red-orange pigment, is the major carotenoid, as many studies report. Other carotenoids from rowanberries are mainly cryptoxanthin [40], but also zeaxanthin, β-cryptoxanthin, all-trans-carotene and γ-carotene [30,41,42], with a high bioaccessibility up to 15.3% [30].

The bright red fruits are also known for their high potassium, calcium and phosphorus content, although differences in concentration were reported due to the growing region and climate condition [43]. Some authors underline the fact that the color clues of the fruits, pH, organic acids, specific sugar, total content of phenolic, vitamin C and antioxidant activity of rowan fruits are significantly different among genotypes and the harsh medium from where they come [4].

In the last few years, the phenolic compound has drawn attention, because the diet intake of phenolic compounds are associated with the prevention of chronic and degenerative diseases, which constitute major causes of death and incapacity in developed countries, such as cardiovascular diseases, type II diabetes, osteoporosis, some types of cancer and neurodegenerative disorders, such as Alzheimer’s disease and Parkinson’s disease. Presently, it is considered that these compounds contribute, at least partially, to the protective effects arising from diets rich in vegetables and fruits, so the study of their role in human nutrition has become a central problem to investigate in food research [44,45].

The content of phenolic compounds in rowanberries is shown in Table 2.

After the analysis of harvested fruits in the Balkan Peninsula, it was discovered that of the phenolic acids, it was mostly the chlorogenic and neochlorogenic acids that were found in higher quantities in fruits and rowanberries pomace [9,46], regardless of the growing area [34]. These acids represent 56–80% of the total phenols found in rowanberries. The content of chlorogenic acids is comparable to that of Arabica coffee beans, the richest source of known phenolic acids at 0.20 mg/g [2,12].

Some authors showed that the main constituents of rowanberry extracts were proanthocyanidins. An aqueous extract of rowanberry pomace has a proanthocyanidins content of approximately 301 mg/g [9]. A study shows that quercetin3-O-(6″-malonil)-glucoside stands for more than 50% of the flavonoid glycosides present in an *S. aucuparia* fruit extract [52].

Eight flavonol glycosides were detected in rowanberry juice—six of which were quercetin: quercetin 3-O-galactozid, quercetin 3-O-glucozid, two quercetin 3-O-dihexozide, quercetin pentose-hexoside and quercetin-pentose hexose—and two glycosides, kaempferol dihexosides. The total content of the identified flavonols in rowanberry juice was 291 mg L^−1^. The same authors show that, from a nutritional point of view, rowanberry juice gives 70 mg of flavonols and 196 mg of chlorogenic acids per serving [13].

Also, the following were detected in rowanberry: among phenolic acids—syringic acid, among flavonols—isorhamnetin, and from the terpenoid class: ursolic acid, squalene, β-amyrin, α-amyrin, cycloartenol, betulin and oleanolic acid [2,18,53].

Aucuparin is the most common biphenyl phytoalexin in *Pyrinae rosaceae* subspecies, which includes economically important fruit-bearing trees like apple trees and pear trees [54]. Phytoalexins are phytochemical substances with antimicrobial and antioxidant properties synthesized by the plants. The following phytoconstituents from the category of phytoalexins were identified in rowanberry fruits: aucuparin, 20-methoxyaucuparin, 40-methoxyaucuparin, noraucuparin, isoaucuparin and 20-hydroxyaucuparin [18].

In general, the results suggest that not only are the fruits and juice a promising source of natural compounds with antioxidant and biological activities, but so too is the rowanberry pomace [9].

## 5. Biopharmaceutical Properties of Rowanberries

Numerous studies show that phytochemical substances from forest fruits, especially phenolic compounds, prove to have a wide range of biological effects, such as antioxidant, anti-inflammatory, antidiabetic, antidiarrheal, antitumoral and also diuretic, vasodilatory and cell-regulating effects [4,9].

In the case of rowanberries, one can assume that chlorogenic acids, which represent approximately 80% of phenolic compounds, are responsible for the health effects of fruits [50]. Chlorogenic acid, first isolated from coffee beans where it has the highest known concentration, is a well-known free radical scavenger, and therefore, plays a protector role in pathologies linked with reactive oxygen species, including cardiovascular disorders, neurological diseases and cancer. Likewise, it is capable of improving glucose tolerance and promoting weight loss [55].

Chlorogenic acids, studied vastly as phytochemicals, can reduce the risk of type 2 diabetes and cardiovascular complications. During the body’s metabolism, one can assume vasodilator substances occur, which lead to a drop in blood pressure [56].

Contrary to isolated or synthetic chemical compounds, vegetal origin materials contain specialized metabolite bodies which, when presented in extracts, exercise health or pharmacology promotion effects and also synergistic and additive effects [57]. 

There is a series of research linked to the effects of rowanberry fruit extracts, its pomace and the plant’s other parts on various diseases or pathogen agents [2,18].

The biopharmaceutical effects of rowanberry are shown in Figure 3.

### 5.1. Antioxidant Effect

Exogenous antioxidants, such as phenolic compounds that can eliminate reactive species, can provoke chelating and can regulate enzymatic and non-enzymatic systems. They are also considered promising protective agents and even therapeutical for oxidative-stress mediated-pathology management.

Free radicals are chemical species that have an unpaired electron, being highly reactive species that can generate or propagate damaging chain reactions.

The mechanisms of action of antioxidants are very complicated but, in general, they manifest in the following directions: (a) through their reducing action, antioxidants can inhibit the formation of free radicals; (b) antioxidants can capture free radicals, acting as a radical scavenger, and by donating an electron or H atom, antioxidants can become an antioxidant radical species that is stable enough to stop the chain reaction; (c) they can create complex transition metal ions, which catalyze prooxidative processes [58,59]. Metal chelators prevent oxidation by inhibiting the redox cycling of metals and producing insoluble metal compounds. The typical metal ions they bind are iron, copper, cobalt, manganese and chromium, which are active pro-oxidants [59].

Considering the mechanism of action, it can be considered that antioxidants have an important role in both preventing some diseases and in reducing their effects [58].

The antioxidant activity of different extracts can be determined by various methods (spectrometry, electrochemical tests, chromatography), which measure either hydrogen atom transfer or electron transfer. These methods provide information on antioxidant potential, but do not faithfully reflect antioxidant activity in vivo. That is why it is recommended that the antioxidant activity of a product be tested using several methods, though these methods are difficult to compare with each other [60]. The results are expressed by reference to the activity of an antioxidant, such as Trolox, in units called the Trolox equivalent, for example, mmoliTE/g, or as percentages of free radical inhibition, measured by the loss of fluorescence or discoloration of the samples.

Table 3 shows ranges of values for the antioxidant activity of rowanberry extracts, determined by different methods, compared to the antioxidant activity of wild bilberries (*Vaccinium myrtillus* L.), which are fruits recognized as having one of the highest antioxidant potentials [61].

The antioxidant activity of rowanberries shows high levels that can be correlated with the polyphenol content and also with a relatively high content of carotenoids.

The antioxidant activity depends on the temperature and storage conditions. A study results show that a temperature of 4 °C was optimal for maintaining antioxidant activity. At −20 °C, the antioxidant activity decreases by approximately a third, which means that the fruits freezing decreases the antioxidant activity. Also, a decreasing, less significant activity occurs at over 25–30 °C [31]. Still, the authors show the antioxidant activity presented a high stability when the extract was subjected to different thermal treatments, pHs and ionic forces, data that confirm the technological potential of this traditional species [30].

Recent studies show the direct connection between the antioxidant activity of various plant extracts and the antitumor [68,69], antidiabetic [70,71,72] and antibacterial [73,74,75] activities.

The reactive oxygen and nitrogen species’ inhibition effects of extracts from various fruits was studied. Although rowanberry extract was weaker than the rest, it still inhibited over 50% ROS production in phagocytes [47].

Another study shows that rowanberry extract significantly inhibits the formation of advanced glycation final products, neutralizing in vivo generated multiple oxidants, increasing human plasma non-enzymatic antioxidant capacity and protecting plasma components (proteins and fat) against oxidative/nitrate damages on significant levels. The fruits’ biological activity is explained through the additive and synergic effects that occur [8].

In another study, using rat livers, it was shown that *Sorbus aucuparia* fruit methanolic extract moderately prevented fat peroxidation (8.21%) [43], while other authors reported that the phenolic substances from wild rowanberries were very active in fat oxidation inhibition, in liposomes and emulsions [50].

Another important role of antioxidants is to protect foods and other oxidation-sensible products during processing and storage in order to prolong their viability terms and to improve quality and safety [51]. In comparison with pure synthetic compounds, natural preparations using phenolic antioxidants can be more effective due to the synergic effects of the different molecules present in plant-based products. Furthermore, the natural ingredients are usually safer than their synthetic counterparts and are therefore preferred by the consumers [76].

Research shows that phenolic extracts from *Sorbus aucuparia* can protect oils from thermal and oxidative degradation during frying. The extracts have a considerable potential as natural antioxidants for polyunsaturated vegetable oils [77].

### 5.2. Antidiabetic Effect

Different rowan species were traditionally used as antidiabetic plants, and several studies confirm this effect [76].

Some polyphenolic compounds regulate blood glucose metabolism by enhancing cellular insulin resistance. Their antioxidant nature and metal chelating activity [78], the ability to trap intermediate dicarbonyl compounds, could be possible mechanisms against glycation and the formation of advanced glycation products [79].

It is known that hyperglycemia is a promoter of oxidative stress by increasing the scavenging of reactive oxygen species and is caused by glucose autoxidation, glucotoxicity and oxidative phosphorylation [80]. Moreover, the insulin resistance that characterizes the diabetic patient causes oxidative stress, and the increased amount of free fatty acids is also involved in the increased promotion of reactive oxygen species, which directly participate in the destruction of the pancreatic cell [80]. This explains the role antioxidants can have in improving diabetic complications. Proanthocyanidins, found in large quantities in rowanberries, have antioxidant and anti-inflammatory properties and prove to be promising substrates to improve insulin sensibility and glucose homeostasis [9].

More so, rowanberries contain sorbitol, which is a suitable sweetener for diabetics [16].

A study reported that [81] *S. aucuparia* fruits inhibited α-glucosidase and were as efficient as an acarbose pharmaceutical inhibitor in maintaining postprandial glycemic control in type 2 diabetes. Other authors [76] have suggested that both flavonoids and terpenoids from rowan fruits can have a beneficial effect in the treatment of the symptoms of type 2 diabetes.

Rowanberry extracts can counteract some side effects of type 2 diabetes, such as cardiovascular complications. One study result showed that extracts from rowanberries had an antithrombotic and protector effect on endothelial functions by protecting human fibrinogen against oxidative modifications, inhibiting thrombin enzymatic properties, diluting the generated fibrin clot and by the inhibition of hyaluronidase activity [82].

In patients with diabetes, there was an increase in the concentration of lipid peroxides in the plasma and also a low level of vitamins C and E [58].

Lipid peroxidation plays a major role in atherosclerosis and in triggering cardiovascular diseases. The polyunsaturated fatty acids in the cell membranes can be oxidized with the formation of free radicals, which are responsible for the toxic processes that take place in the artery walls [58].

The dietary intervention of certain antioxidants, particularly natural ones, can prevent or delay the oxidation of “low-density lipoproteins”, thus reducing the incidence of cardiovascular diseases [58].

At the brain level, there is a large amount of polyunsaturated fatty acids, and the antioxidant systems are low in number [58].

### 5.3. Antimicrobial Effects

The investigation of new plant-based agents could provide alternative antibiotics, potentially combating the development of antibiotic resistance [83].

The antimicrobial effect of rowanberry fruits, pomace and jam was analyzed the most. The results are different and sometimes contradictory because the extract activity depends on the solvent used and the dosage. Additionally, differences in the fruits harvested in different areas can contribute to these variations.

The results of the studies show that rowanberry extracts inhibit the growth of both Gram-positive and Gram-negative bacteria, but Gram-positive bacteria are slightly more sensitive than Gram-negative, a fact explained by the structure of the outer membrane of the bacterial cells, which in Gram-negative bacteria has three functional layers, compared to the two functional layers in Gram-positive bacteria [9].

Gram-negative bacteria pose a significant challenge due to their resistance to most antibiotics or detergents. Treatment options for these bacteria are limited [84].

Also, the antimicrobial resistance of bacteria also depends on the biofilm that bacteria create to survive harsh environmental conditions or to resist the host’s immune system; almost all biofilm communities comprise both Gram-positive and Gram-negative bacteria [85].

As an antimicrobial mechanism of action, polyphenols alter microbial cell permeability, inducing damage to cellular components. Moreover, the presence of the hydroxyl group (OH−) in phenolic compounds plays an important role in bacterial cell death by interacting with bacterial cell membranes through hydroxyl group interactions.

The sensitivity to polyphenols is greater for spore-forming bacteria (such as *Bacillus* strains) compared to non-spore-forming ones [59].

The antibacterial mechanisms of quercetin have been reported to include: altering the permeability of bacterial cells; disrupting bacterial cell walls; inhibiting the synthesis of nucleic acids, thereby affecting the production and expression of proproteins; and reducing enzyme activity [86].

A series of studies reported *Sorbus aucuparia* fruits antimicrobial effects.

Regarding the action of rowanberry extracts on Gram-positive bacteria, a study shows that aqueous ethanol extracts (50%) inhibited the growth of *Bacillus cereus* and *Staphylococcus aureus* [87], while in another study, phenolic-rich fractions from wild *Sorbus aucuparia* showed weak but clear bacteriostatic activity against *Staphylococcus aureus* [51].

Studies on the action of the fraction rich in polyphenols against *Clostridium perfringens*, *Bacillus cereus*, *Staphylococcus aureus* and *Candida albicans* showed the presence of antibacterial activity against all strains tested, except *Bacillus cereus* [55], while the antimicrobial effect of the phenolic extract of rowanberries (1 mg/mL) is very strong against *Bacillus cereus*, weak against *Staphylococcus aureus* and does not affect *Clostridium perfringens* and *Candida albicans* [82]. In another study, *Bacillus cereus*, *Bacillus subtilis* and *Enterococcus faecalis* were the most sensitive of Gram-positive bacteria to rowanberry pomace extracts, while *Listeria monocytogenes* was less sensitive [47].

Some authors report that rowanberry extracts had no activity against the Gram-positive microorganisms *Staphylococcus aureus*, *Enterococcus faecalis* and *Bacillus subtilis*, but showed antimicrobial activity against Gram-negative microorganisms, such as *Escherichia coli* (6/10 compared to ampicillin) and *Pseudomonas aeruginosa* (10/18 compared to ampicillin) [88].

Some authors show that against *Pseudomonas aeruginosa*, only fresh fruit extracts were active, but had no effect on *Escherichia coli* [87].

In another study, *Sorbus aucuparia* fruit extracts (water and methanol) moderately affected the growth of *Escherichia coli* [18,34]. In addition, extracts from wild *Sorbus aucuparia* impaired hemagglutinin-mediated hemagglutination of *Escherichia coli* at concentrations of 1–2 μg and 0.5–1 μg total phenolics/mL, respectively [51].

The fraction rich in polyphenols showed the presence of antibacterial activity against *Campylobacter jejuni* [55], while the phenolic extract of rowanberries (1 mg/mL) is weak against *Campylobacter jejuni*. [82].

In a study on multiple berries, Denev and his collaborators show that extracts of rowanberries, chokeberries and black currants have the most significant microbial activities against a broad spectrum of microorganisms. Rowanberry extract showed the highest activity against *Salmonella enterica* and *Pseudomonas aeruginosa*, moderate activity against *Pseudomonas vulgaris* and moderate-to-weak activity against *Escherichia coli* and *Klebsiella pneumoniae* [47]. Also, rowanberry extract also has significant mitogenic activity [47].

Other authors analyzed the antibacterial activity of fruit juices, as well as water and methanol extracts of pomace from various berries against *Escherichia coli* and *Serratia marcescens*. In general, *Sorbus aucuparia* together with *Ribes nigrum* and *Cornus mas* showed the best ability to inhibit the growth of these germs [89].

The study on rowanberry pomace extracts shows that *Pseudomonas aeruginosa* and *Citrobacter freundii* showed the highest sensitivity among Gram-negative bacteria, while *Escherichia coli* and *Pseudomonas fluorescensis* were the least sensitive [9]. The study also shows the dependence of the activity of the extracts on the different strains, on the solvent used, namely acetone, water and alcohol, but also on the concentration [9]. Acetone extract was the most potent antimicrobial agent, followed by water and ethanol extracts.

The results of studies on the antimicrobial effects of *S. aucuparia* fruits are centralized in the Table 4.

Phytoalexins are considered substances with antifungal action, but they can also have antibacterial activity, resulting in both cell death and non-lethal inhibitions [90]. They do not have a specific structure or chemical character. They are compounds quickly synthesized by plants, defensively, when the plant is infected with a pathogen, at the sites of infection. They have a protective role by inhibiting the maturation of the pathogen, disrupting its metabolism, preventing reproduction or perforating the cell wall of the pathogen in question [91,92].

Aucuparin (2,6-dimethoxy-4-phenylphenol) and its derivatives belong to the biphenyl phytoalexin group and are present in *S. aucuparia*. There are studies on the antifungal effects of such compounds in plants. Aucuparin and noraucuparin isolated from “Florina” apple plantations showed a significant inhibitory effect on the germination of the fungus *Venturia inaequalis* [93]. Of several biphenyls and dibenzofurans tested for their antibacterial activity against some strains of *Erwinia amylovora*, 3,5-dihydroxybiphenyl was the most effective [94].

Eight natural biphenyl-type phytoalexins isolated from the leaves of *Sorbus pohuashanensis*, invaded by *Alternaria tenuissi*, showed significant activity against four other crop pathogens, *Exserohilum turcicum*, *Fusarium graminearum*, *Sclerotinia sclerotiorum* and *Helminthosporium maydis* [95].

Subsequently, the activity of aucuparin and other diphenyl derivatives was evaluated as antimicrobial agents against antibiotic-resistant bacteria in human medicine [96,97].

### 5.4. Cytotoxic Effects

Despite diagnosis and therapeutical progresses, the cancer burden is still heavy worldwide. Nowadays, the toxicity of chemotherapeutic agents for normal cells and the resistance of tumoral cells against this therapy underlines the urgent need for new drugs with minimal side effects. The potential use of natural anticancer agents is entering the cancer research field, and efforts are being made to isolate bioactive products from medicinal plants [19].

Various studies highlight the antitumor effects of polyphenols, both in the treatment of cancer and in the protection and prevention of the appearance of tumors.

Regarding cancer treatment, numerous studies worldwide have shown that polyphenols exhibit antitumor activity through different mechanisms, such as: inhibition of tumor formation, inhibition of tumor progression, induction of apoptosis in cells [98,99,100,101,102,103], stiffening the membranes of cancer cells [104] and modulating the pathways of signaling associated with cell survival [105].

Polyphenols play a crucial role in cancer prevention due to their antioxidant, anti-inflammatory, and immune-modulating properties [105,106]. Additionally, they promote the regulation of intestinal flora [107], which is essential for the body’s immunity.

Cancer cells arise due to genetic mutations in suppressor genes or oncogenes [108]. Polyphenols, by annihilating free radicals, protect the genetic material of the cell from carcinogenesis or other damage [101]. They also help reverse epigenetic changes associated with tumor suppressor gene inactivation [109], which is beneficial in preventing cancer.

Moreover, polyphenols have a synergistic action with existing anticancer drugs, potentiating their therapeutic activity [110,111] and reducing cell resistance to drugs [112]. Due to their properties, plant phytochemicals can be used as adjunctive therapy, reducing the side effects [113,114] of drugs, chemotherapy or radiotherapy [115].

The cytotoxic activity of *Sorbus aucuparia* extracts, in vitro and in vivo, is quite well documented, and these activities are indicated by tumor cell membrane integrity, mitochondrial membrane potential and nuclear size [18].

In a study on the effect of some Siberian wild fruits on prostate cancer, it was found that mountain Siberian ash has significant anticancer activity, provoking the death of up to 90% of cancerous cells, depending on the extract concentration [32].

Some researchers evaluated the cytotoxicity of different agents regarding a series of cancerous cells: *HepG2*, *Caco-2*, *A549*, *HMEC-1* and *3T3* and concluded that *Sorbus aucuparia* fruits extract exerted a relatively high toxicity [87].

Other authors showed that both ethanol extract and watery one showed a significant toxicity, depending on concentration and dosage, against Caco-2 cells [9].

In another study, a rich polyphenols extract (50 µg GAE/mL) from rowanberries reduced HeLa cancerous cells viability by approximately 50% [89], a result confirmed by other authors, with the mention that the cytotoxic activity depended on the used solvent: extracts with methylene chloride were active, while water and methanol extracts presented a weak activity or showed no activity [33,116,117]. Likewise, the fruits’ confectionary was inactive [33].

Studies conducted in 2017 and 2018 showed a potential antitumoral activity in acidified ethanol extract (95%) from rowan fruits, when administered to female mice with Lewis pulmonary carcinoma [118].

In general, it is shown that acetone extracts that contain the highest chlorogenic and non-chlorogenic acids and aqueous extracts which contain the highest quantity of proanthocyanidins show a higher cytotoxicity than ethanol extracts from rowanberry pomace [9].

Another study demonstrates both the initial antitumoral activity *Fructus Sorbi aucupariae* extract saturated with anthocyanin and its enhanced antimetastatic activity when used in conjunction with cyclophosphamide treatment on pulmonary carcinoma mice and B-16 melanoma [119].

The results of the studies on the cytotoxic effects of *S. aucuparia* fruit extracts are centralized in the Table 5.

Other authors show that the anthocyanin content complex from *Sorbus aucuparia* acted on the main indicators of blood erythropoiesis from the mice blood and bone marrow with Lewis pulmonary carcinoma on the background of taking doxorubicin and prevented anemia syndrome development through promoting erythropoiesis regeneration, after its exhausting due to cytostatic agent administration [120]. Likewise, the complex reduces the genotoxic effect of doxorubicin in metaphase plaques of cells bone marrow 24 h, 48 h and 10 days after cytostatic administration. The medium number of individual fragments and cell fraction with gaps and aberrant metaphases also decreases [121].

The effect of fruit extract in various doses was studied in lymphocyte raising on hamsters, and the conclusion was that the lymphocyte proliferation in the treated cells was several times higher than in witnessed untreated cells for the rowan extract, only with the higher dosage used [47].

## 6. Applications

### 6.1. Aucuparin Biosynthesis

Phytochemical substances are compounds that are produced in a plant’s secondary metabolism, in specialized cells, but are not directly involved in primary photosynthetic or respiratory metabolism. They are essential for the survival of plants in their environment and can function as defensive systems (against viruses, herbivores, germs, viruses or competing plants), as signaling compounds (to attract pollinators or animals to disperse seeds) or have a role in protecting the plant from ultraviolet radiation and oxidants. Therefore, secondary metabolites represents adaptive traits subjected to natural selection [122].

Various studies suggest that plants are stimulated to produce more phytochemicals if the environment conditions are more demanding. For instance, a study on *Sorbus* species from the American continent shows that there is greater expression of secondary metabolite genes, a higher antioxidant capacity and a bigger phenol compounds content in samples harvested from higher latitudes in comparison with those from lower latitudes where the climate is milder [76].

In this regard, *Sorbus aucuparia* cells cultures were treated with yeast extract as a stress agent, and it induced aucuparin formation or biphenyl-aucuparin as a major phytoalexin [123,124,125,126].

Furthermore, some research shows that the plants have a memory function for the environmental stress they suffered. When they are subjected to repeated environmental stress they can better and rapidly activate the response and adaptation mechanism to environmental stress, respectively, the production of secondary metabolites [127]. The results of a study showed that biomass accumulation levels and other secondary metabolites in *Sorbus aucuparia* cells cultures subjected to stress repeatedly are significantly higher in certain moments compared to unique stress. More so, it appears that this stress memory mechanism is transmitted over generations in medicinal plants [127].

### 6.2. Nanoparticles Production

Some researchers, taking into account the potential health benefits of rowanberry extract have used it for its unexplored potential in reducing gold and silver salt to form gold and silver nanoparticles [16]. Gold and silver nanoparticles, extremely mono disperse, stable and biocompatible, were successfully synthesized using a simple, efficient and economic method by using an aqueous extract of rowanberries.

The obtained results using different techniques showed that biological compounds from rowanberry extract played a key role in reducing and completely stabilizing nanoparticles [16].

The aqueous extract from rowanberry leaves was successfully used as a reducing agent for silver and gold nanoparticle synthesis from their salty solutions [128]. 

### 6.3. Nutritional and Pharmaceutical Applications of Rowanberry

One can notice that rowanberries are a rich source of vitamin C: 0.1 mg/g in fruits and 0.42 mg/g in jam [33]. It was reported that rowanberries contain a quantity of ascorbic acid three times higher than oranges [2].

The recommended diet dosage of ascorbic acid is 60 mg per day, while only 5–7 mg per day prevents scurvy [2]. Vitamin C deficiency results in impaired immunity and greater susceptibility to infection [129].

Vitamin C is a powerful antioxidant and contributes to immune defense by supporting the various cellular functions of both the innate and adaptive immune systems. It also has a gene regulation effect. Its physiological role is vast and encompasses very different processes, from facilitating iron absorption to involvement in hormone and carnitine synthesis and important roles in epigenetic processes [130].

Some studies show that vitamin C prevents conditions related to metabolic syndrome and improves the condition of patients diagnosed with this syndrome [131].

Other studies highlight the role of vitamin C in the prevention and treatment of gout, indicating a significant correlation between high vitamin C intake and lower serum uric acid levels [132]. Vitamin C intake is associated with a reduced risk of periodontal disease [133] but also a lower risk of mortality from various causes, such as various types of cancer and cardiovascular diseases [134].

Supplementing the diet with vitamin C with an intake of 100–200 mg/day prevents and treats respiratory and systemic infections [129]. It has been extensively used in the treatment of patients with SARS-CoV-2 [135].

Vitamin C is used in high doses in the case of cancer patients undergoing chemotherapy, because they are deficient in vitamin C, with intravenous administration being more effective [136,137,138].

Vitamin E is also present in a different tocopherol form. The recommended intake of vitamin E for adults is in the range of 7 to 15 mg per day [2]. The epidemiological studies showed that humans who consumed vitamin E richer foods had a lower incidence of cancer, dementia and/or cardiovascular diseases [39].

Different precursors of vitamin A are present in rowanberries. Their action, like other phytoconstituents, depends on the body’s ability to retain them. Bioaccessibility describes the ingested compound quantity which is released into the dietary matrix during the digestive process and becomes available for intestinal absorption [139]. Processing operations, especially drying and grounding, reduce the particles’ size, which favors carotenoids release, therefore increasing their bioaccessibility. The positive effect of food processing on carotenoid bioaccessibility correlates positively with in vivo studies on carotenoid bioavailability, confirming the fact that processed vegetal food consumption improves carotenoid intake [140]. The necessary medium intake is of 2–7 mg per day.

Higher intake of vitamin C and a higher concentration of vitamin E and β-carotene are associated with a lower risk of cardiovascular mortality [141].

Rowanberries are also rich in phenolic compounds. Among them, chlorogenic acids stand out in large quantities. Various studies show that chlorogenic acid can play an important role in promoting human health. It has antioxidant, anti-inflammatory, antibacterial and antiviral effects. It is hypoglycemic and lipid-lowering, modulating glucose and lipid metabolism in both genetic and metabolic disorders [141,142,143,144]. It protects and improves the functioning of the cardiovascular system and is useful in hypertension [141]. It provides liver, gastrointestinal and renal protection [145,146,147,148]. It protects and stimulates the central nervous system and can regulate learning, memory, cognitive ability, alleviate anxiety, depression and other symptoms of post-traumatic stress disorder [149]. It has antimutagenic, anticarcinogenic and immunomodulatory properties, and is able to play an important role in the treatment of cancer [142,145,146,147,148].

Several in vivo and in vitro studies have demonstrated that chlorogenic acid can protect against the toxicity induced by chemicals of various classes, such as fungal/bacterial toxins, pharmaceuticals, metals, pesticides, etc. [150].

Quercetin is also found in large quantities in rowanberries. There are numerous studies on the effects of quercetin in the prevention and treatment of many diseases. It has psychostimulant effects and improves mental and physical performance. Quercetin may have an important role in the prevention and treatment of aging-related brain disorders and several degenerative diseases, including Parkinson’s disease and Alzheimer’s disease, by improving neurogenesis and neuronal longevity [151,152,153]; side effects are insignificant or absent.

It is a powerful antioxidant, inhibiting lipid peroxidation, platelet aggregation and stimulates mitochondrial biogenesis [154,155]. Like other flavonoids, it prevents platelet aggregation and atherosclerotic plaque formation and promotes the relaxation of cardiovascular smooth muscles. It has antiarrhythmic effects and is a good antihypertensive [156], but it is also beneficial in relieving hypotension [157].

Quercetin stands out for its anti-inflammatory activity [158] and ameliorating the complications of diabetes [159,160]. It has proven carcinostatic effects [156,161].

Quercetin reduces the risk of infection and has proven antiviral properties against a large number of influenza virus strains [162], *herpesvirus, cytomegalovirus* and *varicella-zoster virus* [163], and it may also be beneficial in the treatment of AIDS [156]. Recent studies demonstrate the anti-coronavirus effect of quercetin [164,165,166,167,168].

The antiviral effects of quercetin are amplified by the presence of vitamin C, with a synergistic effect, due to the overlap of antiviral and immunomodulatory properties and the ability of ascorbate to recycle quercetin, increasing its effectiveness, both for prophylaxis and for the treatment of patients with COVID-19, as adjuvants of antiviral drugs [156].

A study shows that aucuparin inhibits pulmonary fibrosis through its anti-inflammatory activity and sustains its potential of being a therapeutical drug in idiopathic pulmonary fibrosis treatment—a pulmonary disorder which results in lung scarring from unknown reasons and of which no precise treatments have been identified [169].

The effects of the *S. aucuparia* phytotherapeutic phytoconstituents are summarized in Table 6.

### 6.4. Toxicology

Rowanberries must have their toxicity considered prior to use. Two toxic substances were identified in rowanberries [2]. Parasorbic acid was reported in fruits as 4–7 mg/g and in seeds as 0.08–0.12 mg/g of fresh weight [22]. Ingested in large quantities, this compound can the irritate gastric mucosa, can produce indigestion and can even affect the kidneys.

Through heating or frosting, p-sorbic acid isomerized in sorbic acid is nontoxic. Some authors recommend harvesting the fruits after the first frost [43].

The other toxic compound identified in rowanberry seeds is cyanogenic glycosides (prunasin), which can release hydrocyanic acid, which in certain concentrations can cause respiratory failure and even death. For safety, the authors recommend removing the seeds before using the fruits [2].

Figure 4 and Figure 5 show the structure of parasorbic acid and prunasin, respectively [170].

## 7. Conclusions

Widespread throughout the Northern Hemisphere, rowanberries can be an abundant source of compounds with proven biological properties. Its fruits are rich in vitamins, especially vitamins C, E and provitamin A, and important minerals such as iron, copper, zinc, potassium and magnesium.

The polyphenols found in large quantities in rowanberries also explain the medicinal properties of the fruit, known in ethnomedicine. Biphenyl phytoalexins are also a category of compounds with antifungal and antimicrobial action. Adding to the wealth of phytoconstituents is their synergistic action in the prophylaxis, prevention and treatment of many ailments of the modern world. More and more studies show that rowanberry extracts exhibit potent antioxidant properties, antidiabetic effects and antimicrobial effects, are particularly effective against Gram-negative bacteria and have important cytostatic effects, enhancing conventional anticancer treatments while reducing their side effects. Furthermore, rowanberries are viable candidates in nanoparticle synthesis and aucuparin biosynthesis.

Studies are needed on the effects of rowanberries in the prevention and treatment of other conditions, in which polyphenols have shown clear effects, such as the age-related damage, cognitive decline and degenerative diseases, including Alzheimer’s disease and Parkinson’s disease, cardiovascular diseases, which along with degenerative ones are the main causes of death worldwide and last but not least, antiviral activity, in the context of the emergence of new dangerous strains, as witnessed during the SARS-CoV-2 pandemic. The results on phytochemicals, antioxidant potential and other bioactivities of *Sorbus aucuparia* extracts are promising. They reveal the prospects of its use in dietary supplements, cosmetics and functional foods, with valuable biological effects for health.

## Figures and Tables

**Figure 1 plants-12-03225-f001:**
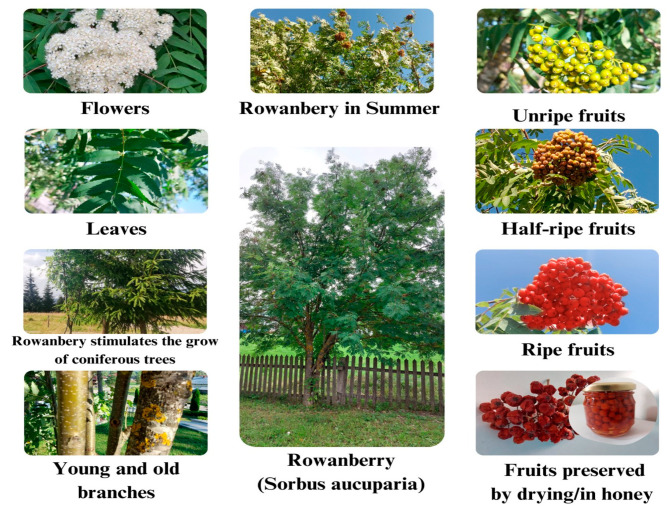
The rowanberry (*Sorbus aucuparia*)—aspects.

**Figure 2 plants-12-03225-f002:**
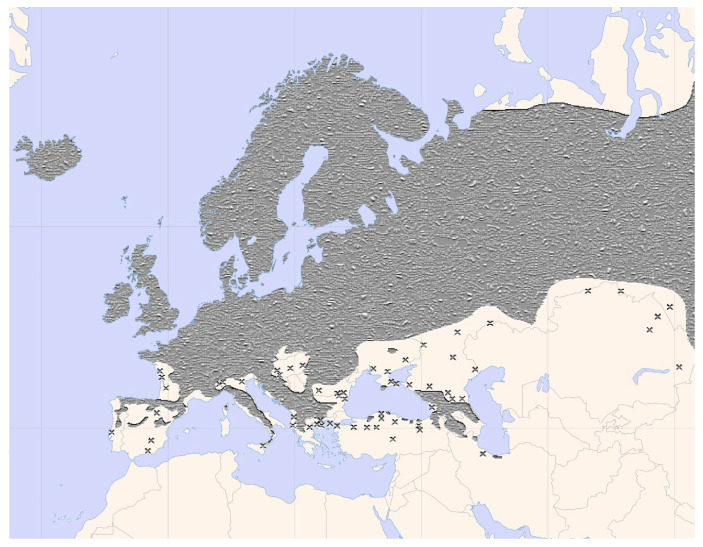
The spread of *Sorbus aucuparia* in Europe. The gray color represents abundant spread. ”x” represents limited distribution.

**Figure 3 plants-12-03225-f003:**
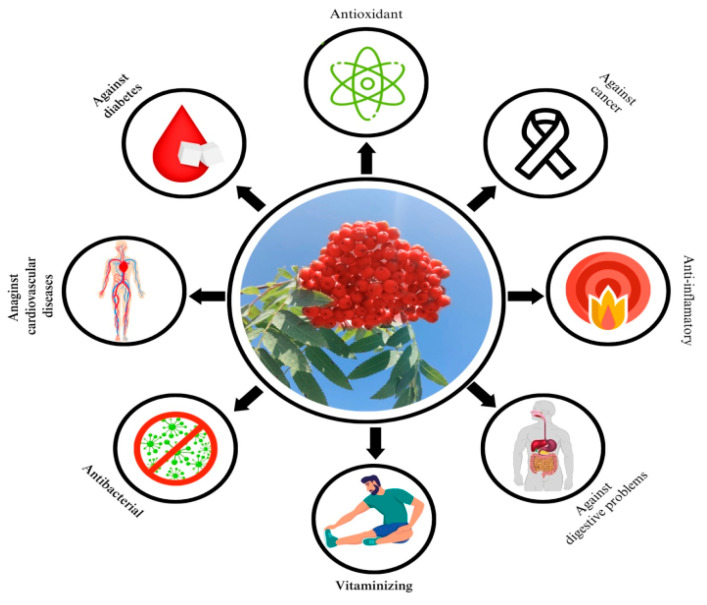
Biopharmaceutical effects of rowanberry (*Sorbus aucuparia*).

**Figure 4 plants-12-03225-f004:**
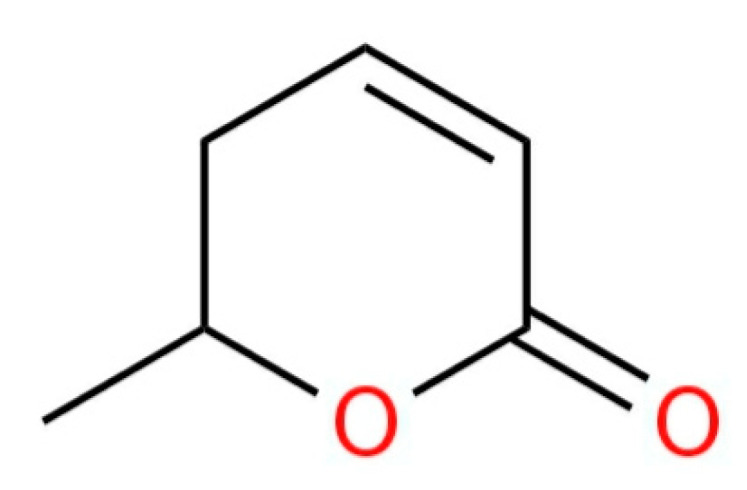
Structure of parasorbic acid.

**Figure 5 plants-12-03225-f005:**
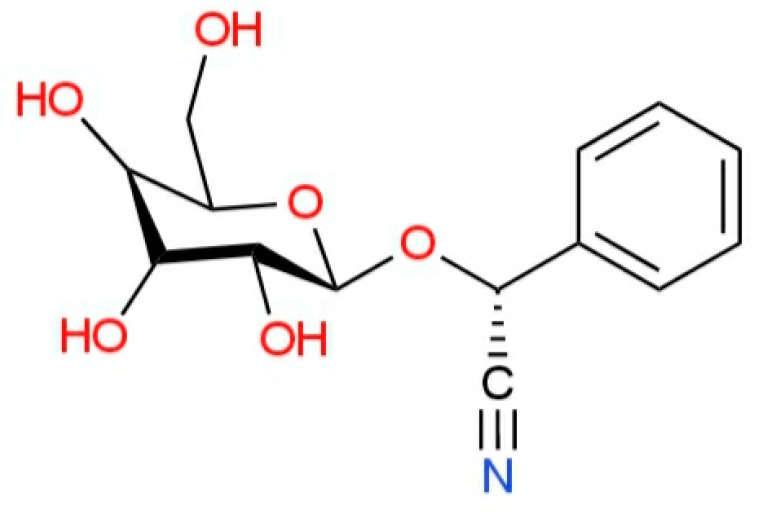
Structure of prunasin.

**Table 1 plants-12-03225-t001:** Nutritional and phytotherapeutic principles found in 100 g of dried rowanberries.

Phytotherapheutic Phytoconstituents	Quantity %	Source
Total sugars	8.73–21.69	[11,27,28]
Glucose	3.76–5.29	[11,27,28]
Fructose	2.48–4.16	[11,27,28]
Sucrose	0.17–0.81	[11,27,28]
Sorbitol	2.68–13.41	[11,28,29]
Organic acids	0.2566	[11]
Acetic acid	0.008	
Citric acid	0.12–10.9	[11,27,28,30,31]
Fumaric acid	0–0.0016	[11,27,28]
Malic acid	2.20–3.34	[11,27,28,30,31]
Oxalic acid	0.38	[27]
Shikimic acid	0.0028–0.003	[11,20]
Sorbic acid	0.75	[27]
Tartric acid	0.01–0.03	[11,27,28]
Total minerals	8–9	[27]
	Quantity mg/100 g	
Al	0.0041	[27]
Ca	0.249–29.9	[2,27]
Cl	<0.03	[27]
Cu	0.0004–0.294	[2,27,32]
Fe	0.0079–2.42	[2,27]
K	1.288–154	[2,27]
Mg	0.095–27.84	[2,27]
Mn	0.0062–0.503	[2,27]
Na	0.0056	[27]
P	0.139–123	[2,27]
S	0.053	[27]
Si	0.0233	[27]
Zn	0.0026–0.861	[2,27,32]
Vitamin C	10–42	[27,30,33]
Vitamin E	0.423–0.718	[34,35,36]
α-Tocopherol	0.34–0.489	[34,35,36]
γ-Tocopherol	0.025–0.171	[34,35,36]
δ-Tocopherol	0.058	[35,36]
Carotenoids	2.5–21.65	[30,37]
Carotenes	0.098–0.1	[30,34]
Cis-carotenes	0.15	[30]
All-trans-carotenes	1.78	[30]
Cryptoxanthin	1.37	[30]
Lutein	0.013	[30]
Lycopene	0.007	[34]
Zeaxanthin	0.003–1.11	[30,34]

**Table 2 plants-12-03225-t002:** The content of phenolic compounds in rowanberries.

Compounds	Content in Fruits mg/g dw	Content in Fruits mg/g fw
Totalpolyphenols content mg eqGAE/100 g dw	190–54,000 [8,28,30,32,43,46,47]	
Phenolic acids	0.0002–269.43 [8,37,47,48]	5.25–15.91 [34]
1-caffeoylquinicacid	6.81 [8]	
3-caffeoylquinicacid (neochlorogenic acid)	0.000705–24.65 [4,8,33,47,49,50]	0.0064–0.369 [11,48]
4-caffeoylquinicacid (cryptochlorogenic acid)	8.33 [8]	4.9 [11]
5-caffeoylquinicacid (chlorogenic acid)	0.0004–155.12 [4,8,30,34,47,49,50,51]	0.0029–0.399 [11,48]
Gallic acid	0.039 [30]	
Caffeic acid	0.021–3.46 [8,30]	
Ferulic acid	0.0078–9.59 [30,50]	
p-coumaric acid	0.004 [30]	
Hydroxybenzoic acid	0.107–0.11 [30,50]	
Protocatechuic acid	0.018–2.80 [8,30]	
Flavonoids	0.681–1.84 [37,50]	
Flavonols	1.54 [50]	0.1488 [11]
Quercetin	0.0028–1.06 [4,8,30,34]	0.00051 [48]
Kaempferol		0.006 [48]
Proanthocyanidins	5 × 10^−6^–0.92 [34,37,50]	0.0107 [48]
Flavanols	0.97 [50]	0.2494 [11]
Catechin	0.017–1.3 [4,30]	
Epycatechin	0.033–0.074 [4,30]	0.0191 [11]
Procyanidin B1	0.022–0.085 [4,30]	
Procyanidin B2	0.092 [4]	
Procyanidin C1	0.103 [4]	
Anthocyanins	0.12 [50]	0.1012 [11]
Cyanidin-3-glucoside	0.001 [4]	0.00209 [11]
Cyanidin-3-galactoside	0.183 [4]	0.09912 [11]
Quercetin-3-galactoside (Hyperoside)	0.024–1.36 [8,33,34,49]	
Quercetin-3-rutinoside (Rutin)	4.01 × 10^−5^–0.0006 [33,34,49]	
Quercitin-3-glucoside (Isoquercitin)	0.0061–2.29 [4,8,33,34,49]	0.02612 [11]
Kaempferol—3-O-glucoside	0.0085–0.009 [34]	0.0024 [11]
Quercetin-3-O β-sophoroside	3.57 [8]	
Hydrolyzable tannins		0.0305 [11]

**Table 3 plants-12-03225-t003:** Antioxidant activity of rowanberry extracts compared with that of wild bilberries.

Antioxidant Activity Rowanberries	Ranges of Values	Antioxidant Activity Bilberries
Antioxidant activity IC 50 g/mL	8.93–4260 [33,34,46,48,62]	3.99 [63]
TP FRAP mmol Fe^2+^/kg dry fruits)	315–54,670 [32,33,48,49,62]	73.16 [64]
TP ABTS mmolTE/100 g	0.76–5.84 [30,49]	6–35.34 [65,66]
AA%	65.2–92.2 [46]	81.94–82.38 [63,67]
DPPH Trolox mmol Trolox/kg dry fruits	10.84–7380 [30,32,33,48,62]	5366 [63]
ORAC mmol TE/L	23.689 [47]	50.87 [65]

**Table 4 plants-12-03225-t004:** The antimicrobial effects of *S. aucuparia* fruits.

Extracts	Gram-Positive Bacteria						Gram-Negative Bacteria									Fungi
	B. c.	S. a.	C. p.	B. s.	E. f.	L. m.	E. c.	P. a.	C. j.	S. m.	P. v.	S. e.	K. p.	C. f.	P. f.	C. a.
Aqueos extract	1			1			1			4						
Water–ethanol 50%	3	3					0	3								
Water–methanol		2					2									
Methanol	4	0		0–2	0		1–3	3		1						
Acetone		1				2	1	3			2	3	1			
Phenolic 1 mg·mL	4	1	0						1							0
Rich in polyphenols	0	2	3				3		3							3
Pomace	3			3	3	1	1	3						3	1	
jus	2			1			1			4						

B. c.—*Bacillus cereus*, S. a.—*Staphylococcus aureus*, B. s.—*Bacillus subtilis*, C. p.—*Clostridium perfringens*, E. f.—*Enterococcus faecalis*, L. m.—*Listeria monocytogenes*, E. c.—*Escherichia coli*, P. a.—*Pseudomonas aeruginosa*, C. j.—*Campylobacter jejuni*, S. m.—*Serratia marcenses*, P. v.—*Pseudomonas vulgaris*, S. e.—*Salmonela enterica*, K. p.—*Klebsiella pneumoniae*, C. f.—*Citrobacter freundii*, P. f.—*Pseudomonas fluorescensis*, C. a—*Candida albicans*. 0—no activity, 1—poor activity, 2—moderate activity, 3—strong activity, 4—very strong activity.

**Table 5 plants-12-03225-t005:** The cytotoxic effects of *S. aucuparia* fruit extracts.

Solvents	HeLa	HepG2	Caco-2	A549	HMEC-1	3T3	Lewis Lung Carcinoma	Prostate Cancer
Acetone + water		3	3	3	3	3	3	
Methanol + water	0–1							4
Water, ethanol			3					
Methylene chloride	2–3							

0—no activity, 1—poor activity, 2—moderate activity, 3—strong activity, 4—very strong activity.

**Table 6 plants-12-03225-t006:** The phytotherapeutic effect of phytoconstituents of rowanberries.

Phytotherapeutic Phytoconstituents	Diet Dosage	Effects
Vitamin C	60 mg/day 5–7 mg/day prevents scurvy	Powerful antioxidant Sustains immunity Gene regulation Iron absorption Hormone synthesis Metabolic syndrome—prevention and treatment Gout—prevention and treatment Periodontal disease—prevention Respiratory infections—prevention and treatment Cancer treatment—assists chemotherapy
Vitamin E	15 mg/day	Cancer prevention Dementia prevention Cardiovascular-diseases prevention
Vitamin A	2–7 mg/day	Lower risk of cardiovascular mortality Eyesight enhancer Strengthen the immune system Sustains the bound healt
Chlorogenic acid		Antioxidant Anti-inflammatory Antibacterial Antiviral Hypoglycaemic and lipid lowering Antihypertension Liver, gastrointestinal, renal protection Stimulates nervous system Antimutagenic, anticarcinogenic Protects against chemicals toxicity
Quercetin		Antioxidant Anti-inflammatory Antidiabetic Antiviral Improves mental and physical performance Degenerative diseases—prevention and treatment Prevent atherosclerotic plaque formation Regulating arterial tension, antiarrhythmic Cytotoxic
Aucuparin		Pulmonary fibrosis

## Data Availability

No new data were created in this study.

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
