# Peer review of "Rowanberry—A Source of Bioactive Compounds and Their Biopharmaceutical Properties"

_plants, 2023, doi:10.3390/plants12183225_

Round 1
Reviewer 1 Report
The purpose of a review is to lay out data and information on a particular topic. In essence, to build a picture. Reviews are read to find out what is currently known about a specific topic and to gather information and references for further study and use. You have done this very well. However, there is more that you need to do. Below is my usual lecture on writing reviews, followed by specific comments on your paper
There are 3 important factors: accuracy, data, and presentation. Accuracy is the most difficult one. It takes time and attention to detail. Data needs to be comprehensive, complete, and presented in an orderly fashion. Presentation needs to be interesting. The better the presentation the more often a review is cited. Part of presentation is clear, concise, succinct text. Tables, figures, and charts provide repetition of the data in the text, make a review memorable, lead to more citations than the best text. But your text is better when you use tables, figures, and charts to write the text.
Your figures are excellent. I would suggest you use World flora online to audit your plant taxons where you use them. If there are differences, you can put the World Flora name in parenthesis. And you only have to list the full taxon the first time the plant is mentioned (or do a table for all the ones mentioned). https://www.worldfloraonline.org/
In general, when listing items, it is wise to do it alphabetically unless it is clear what the order is. Table 1 is a good example. The sugars are listed in a known and clear order. The organic acids should be either alphabetically or by number of carbons. For the minerals, alpha is my preference but you could list by groups such as electrolytes, essential, etc. The tocopherols appear to be missing their designation (alpha, beta, etc.) For carotenoids, keep the carotenes together and the rest in alpha order. Your text discusses the differences in quantity and being able to spot the compound easily will be appreciated.
Table 2 needs some work. You should separate the total polyphenol data from the antioxidant data. For the antioxidant data you need to give the control value because the methods are different.
Tables 3 and 4 could be combined. Your header row could be compound, structure, mg/100g DW, mg/100g FW. You could group the compounds by their classes. I am not certain how you can separate the references for DW and FW, but I am certain you are able to do it. Compounds present without amount provided can be listed in a separate table by classes.
You should provide the structures of the toxic compounds.
This is a good paper. More attention to details will make a very excellent paper. I enjoyed reading it.
Lastly, but most important, you should have a native English speaker review your paper. This person does not need to be a scientist. Some sentences in this paper are not readily understood. A librarian can help you find someone.
Reviewer 2 Report
In the MS plants-2551667, the authors claim the readers' attention describing the bioactive constituents and pharmacological properties of Sorbus aucuparia L. The MS has 84 references, 26 published in the last 5 years.
The review is well-documented and structured as follows:
1. Introduction
2. History and origins
3. Botanical description
4. The chemical composition of rowan berries
5. Biopharmaceutical properties of rowan berries
5.1. Antioxidant effect
5.2. Anti-diabetes effect
5.3. Anti-microbial and bacterial effect
5.4. Anti-tumor effect
6. Aucuparin biosynthesis
7. Nanoparticles production
8. Toxicology
9. Conclusions
The following comments and suggestions are available below:
1. Table 1 (line 159) - the authors are encouraged to replace "principles" with phytoconstituents in the table caption and first row.
2. Table 2 shows two classes of phenolic compounds (polyphenols and flavonoids) as total contents and antioxidant activities determined through various methods and differently expressed. The reviewer suggests removing Table 2 and registering contained data at each suitable section.
3. The authors are encouraged to place the ideas from lines 167-174 and 181-188 in a new suggested section, Applications. It could include nanoparticle production and other nutritional and pharmaceutical applications of Rowanberry and/or main bioactive constituents (e.q. phenolic acids, quercetin, etc., incorporated in various pharmaceutical formulations). This section could contain several aspects from Section 8 (Toxicology).
4. In the same section 4 entitled "The chemical composition of rowan berries," the authors could include the distinct metabolites synthesized in special conditions described in current section 6. Aucuparin biosynthesis.
5. Section 5, Biopharmaceutical properties of rowan berries, could be enriched with the presentation of mechanisms of action for each bioactivity, considering that the essential secondary metabolites are phenolic compounds. The authors could relate their antioxidant potential with antimicrobial, cytotoxic, and other effects, supporting their statements with the most recently published studies from the scientific literature.
6. Subsection 5.3. "Anti-microbial and bacterial effect" could be entitled only antimicrobial effects. In the current version, the authors described only the antibacterial effects.
First, they are encouraged to present antibacterial and antibiofilm effects, organized considering the bacterial type (Gram-positive and Gram-negative).
Then, they can discuss the antifungal effects of the biphenyl phytoalexins (Aucuparin).
7. Subsection 5.4. could be entitled Cytotoxic effects, containing data about cytotoxicity of various Rowanberry extracts on tumor and normal cells from in vitro and in vivo studies. On tumor cells, the authors can mention the synergism with anticancer drugs. The anticancer activity could be analyzed considering three aspects: therapeutic, protective, and prophylactic, and Rowanberry can be helpful in multifarious ways.
8. With all suggested aspects, the authors could reformulate the suitable Conclusions, thus offering valuable and most recent information about Rowanberry.
Round 2
Reviewer 1 Report
This Rowanberry paper could be an excellent review. However, it reads as a written lecture. A good review has several accurate tables. Table 2 (marked as table 4) should be reviewed by a chemist as hydrolyzable tannins are not flavonoids and 3 compounds above it and 3 below are flavonol glycosides. The section on antimicrobial activity needs a table which includes species, plant part or preparation, extraction solvent, individual columns for the most often tested bacteria, and column(s) for the rare ones. A few other sections should have tables too. It is not necessary for all sections. Table 3 should mention that bilberries were used as well as blueberries; both are in the Vaccinium genus. Species should be verified on https://www.worldfloraonline.org/. Articles listed are not always cited.
A native English speaker should read the paper. This person does not have to be a scientist. A librarian could help you find someone.
Reviewer 2 Report
The reviewer congratulates the authors for all their efforts to revise the MS plants-2551667 according to the previous review report. The revised version is substantially improved and suitable for publication in Plants MDPI Journal.
